# Real-World Outcomes of First-Line Chemotherapy in Metastatic Pancreatic Cancer: A Nationwide Population-Based Study in Korea

**DOI:** 10.3390/cancers16183173

**Published:** 2024-09-16

**Authors:** Chan Su Park, Byung Kyu Park, Joung-Ho Han, Kyong Joo Lee, Kang Ju Son

**Affiliations:** 1Division of Gastroenterology, Department of Internal Medicine, National Health Insurance Service Ilsan Hospital, Goyang 10444, Republic of Korea; wpakcs@nhimc.or.kr; 2Department of Internal Medicine, Chungbuk National University College of Medicine, Cheongju 28644, Republic of Korea; joungho@cbnu.ac.kr; 3Division of Gastroenterology, Department of Internal Medicine, Hallym University Dongtan Sacred Heart Hospital, Hallym University College of Medicine, Hwaseong 18450, Republic of Korea; kyongjoolee1214@gmail.com; 4Department of Policy Research Affairs, National Health Insurance Service Ilsan Hospital, Goyang 10444, Republic of Korea; sonkangju@nhimc.or.kr

**Keywords:** metastatic pancreatic cancer, chemotherapy, survival, FOLFIRINOX, gemcitabine plus nab-paclitaxel

## Abstract

**Simple Summary:**

This study investigated the nationwide real-world outcomes of chemotherapy in 8651 patients with metastatic pancreatic cancer. Overall survival improved from 2012 to 2019 and was evident after the introduction of gemcitabine plus nab-paclitaxel (GnP) and FOLFIRINOX. Propensity score matching revealed no difference in overall survival between these two regimens. The findings demonstrate that advances in chemotherapy have improved survival outcomes nationally, comparing the effectiveness of GnP and FOLFIRINOX using real-world data.

**Abstract:**

Background/Objectives: This nationwide population-based study investigated the overall survival (OS) of patients with metastatic pancreatic cancer (mPC) receiving first-line chemotherapy. Methods: Data from the National Health Insurance Service linked to the Korea Central Cancer Registry were used. Patients with mPC receiving first-line chemotherapy (2012–2019) were included and followed up until 2020. The gemcitabine plus nab-paclitaxel (GnP) and FOLFIRINOX groups were matched according to age, sex, and comorbidities. Results: In total, 8652 patients with mPC were treated with chemotherapy. GnP and FOLFIRINOX have been administered since 2016 and 2017, respectively. The median OS increased annually from 6 months in 2012–2013 to 10 months in 2018–2019. The median OSs in the GnP and FOLFIRINOX groups were significantly longer than those in patients receiving gemcitabine ± erlotinib. A total of 1134 patients from both the GnP and FOLFIRINOX groups were selected using propensity score matching. Before matching, the median OS was longer in the FOLFIRINOX group than in the GnP group (*p* = 0.0029). After matching, however, there was no significant difference in the median OS between the two groups (11 vs. 11 months, respectively, *p* = 0.2438). Conclusions: Patients with mPC receiving chemotherapy have shown improved OS since the introduction of GnP and FOLFIRINOX. After matching, OS did not differ between the GnP and FOLFIRINOX groups.

## 1. Introduction

Pancreatic cancer (PC) is a lethal disease, with a 5-year survival rate of less than 10% [1,2]. Surgery is the only chance for cure, but most patients are diagnosed with unresectable disease; therefore, palliative chemotherapy remains the mainstay of treatment for most patients.

In 1997, gemcitabine monotherapy was established as the standard first-line palliative chemotherapy for PC [3]. Many cytotoxic agents combined with gemcitabine have been tried; however, they have failed to significantly improve patient overall survival (OS) [4,5,6]. In 2011, FOLFIRINOX (fluorouracil, folinic acid, irinotecan, and oxaliplatin) emerged as the initial treatment regimen, exhibiting superior results when compared to gemcitabine (median OS: 11.1 vs. 6.8 months, respectively) [7]. Subsequently, the combination of gemcitabine and nab-paclitaxel (GnP) was shown to improve survival when compared to gemcitabine monotherapy (median OS: 8.5 vs. 6.7 months, respectively) [8]. These two combination chemotherapies are currently recommended as the first-line treatment for metastatic PC (mPC).

Chemotherapy for PC has advanced; however, large-scale population-based studies of survival improvements with chemotherapy are limited. Two population-based studies conducted in Canada have shown improved survival in patients with PC receiving chemotherapy [9,10]. Recently, a population-based study in Korea has also reported a gradual improvement in the survival of these patients [11].

A direct comparison between GnP and FOLFIRINOX in randomized controlled trials has not been conducted, and the results of retrospective indirect comparative studies are conflicting. Several retrospective studies from single or multiple centers suggest that these two regimens have similar effects on survival [12,13,14,15,16]. Contrarily, there are reports that one regimen, either FOLFIRINOX [17,18] or GnP [19,20], is more effective. Some population-based studies have shown that FOLFIRINOX has survival benefits over GnP [21,22,23,24], whereas others have reported that there is no difference [25,26]. A meta-analysis of first-line chemotherapy for mPC of NALIRIFOX (fluorouracil, leucovorin, liposomal irinotecan and oxaliplatin), FOLFIRINOX and GnP in seven clinical trials was recently reported, but it had limitations in terms of adjustment and population heterogeneity [27].

The actual clinical situation is often different from the environment in which the clinical trial was conducted. Therefore, treatment effects proven through clinical trials may not be equally observed in the general population [28]. Notably, there is a discordance between clinical trial efficacy and the effectiveness of real-world evidence [29,30]. The efficacy of a new cancer treatment tested in clinical trials needs to be validated through a population-based study to determine if the expected results are achievable in the general population [31]. A nationwide population-based study involving all patients with PC across the country may provide real-world evidence for treatments that can be applied in practice. Furthermore, GnP and FOLFIRINOX can be compared by including a sufficient number of patients and minimizing confounding variables through matching.

Therefore, this nationwide population-based study aimed to analyze the OS in patients with mPC receiving first-line chemotherapy in Korea. The outcomes of GnP and FOLFIRINOX were compared using propensity score matching (PSM).

## 2. Materials and Methods

### 2.1. Data Sources

This study used data from the National Health Insurance Service (NHIS) linked to the Korea Central Cancer Registry (KCCR). The NHIS is a mandatory nationwide health insurance system covering more than 98% of the Korean population. The medical information on almost all patients in healthcare institutions is prospectively integrated into the NHIS claims database. This includes extensive information on diagnoses, procedures, comorbidity codes, demographic characteristics, admission and ambulatory care, and medications. The KCCR is a nationwide cancer registry widely used to monitor cancer incidence, mortality, and survival. It was established by the Ministry of Health and Welfare and is operated by the National Cancer Center in Korea. The KCCR database contains cancer information such as Surveillance, Epidemiology, and End Results Program (SEER) staging and histological diagnosis.

The NHIS claims database does not include cancer-related information (e.g., cancer staging). The NHIS–KCCR linked database was, therefore, used to ensure the accuracy of diagnoses and specifically select patients with mPC from all patients. This linked database classifies age into 5-year increments and displays all dates in months rather than days in accordance with a policy to prevent personal identification.

### 2.2. Study Population

This study was conducted among newly diagnosed patients with mPC receiving chemotherapy between 1 January 2012, and 31 December 2019. Patients with PC were defined as those with a PC diagnosis code (C25) and a rare and intractable disease (RID) registration code (V193) in the NHIS between January 2012 and December 2019, and registered with PC in the KCCR between January 2011 and December 2019. The RID registration process ensures the reliability of the cancer diagnosis [32].

Patients diagnosed with a pancreatic neuroendocrine tumor, as well as those under 20 years of age or with unreliable data were excluded from the study. A pancreatic neuroendocrine tumor was defined as a histological diagnosis of a neuroendocrine tumor in the KCCR or as a case with three or more claims of pancreatic neuroendocrine tumor (C254) in the NHIS. After exclusion, only patients with mPC were selected based on the SEER stage recorded in the KCCR. Among these patients, those who underwent pancreatic surgery and those who started chemotherapy after 2019 were excluded. Finally, patients with mPC who received chemotherapy without surgery (2012–2019) were selected for further analysis.

### 2.3. Definitions of Conditions

The month of PC diagnosis was defined as the month in which the diagnosis (C25) was first confirmed. Chemotherapy was defined as a claim for an anticancer drug injection code (KK151–156) or a claim for anticancer drugs within 6 months of PC diagnosis. Cases in which the same drug was claimed at least three times were recognized as chemotherapy. The chemotherapy regimens were classified into gemcitabine monotherapy, gemcitabine plus erlotinib, GnP, FOLFIRINOX, and other drugs. The total number of chemotherapy cycles was calculated by checking the number of times each anticancer drug was prescribed and considering each chemotherapy cycle. The chemotherapy period was defined as the period from the first to the last month of an anticancer drug claim. Pancreatic surgery was defined as a case in which the procedure code for pancreatic surgery was claimed.

### 2.4. Propensity Score Matching between the GnP and FOLFIRINOX Groups

We evaluated the differences in clinical outcomes between patients receiving first-line chemotherapy with GnP or FOLFIRINOX. PSM was used to correct for the baseline differences between the GnP and FOLFIRINOX groups. The two groups were matched (1:1 ratio) based on age, sex, and the Charlson Comorbidity Index (CCI) score. The presence of comorbidities was classified (≤5, 6–9, or ≥10 points) using the CCI score [33], based on a diagnosis confirmed from the NHIS database within 1 year of PC diagnosis.

### 2.5. Study Outcomes

The primary outcome was OS, which was defined as the time from the initial month of first-line chemotherapy to death or the end of follow-up. The follow-up period was until December 2020, or the month of death. The survival time was censored for patients who were alive at the end of the study period.

The secondary outcome was the safety of chemotherapy in patients who received GnP or FOLFIRINOX, assessed through subgroup analyses. In a study using administrative data, safety outcomes can be evaluated with the following conditions: all-cause emergency center visits, all-cause hospitalizations, and febrile neutropenia [22,34]. These conditions were modified and defined according to the medical environment in Korea as follows. (1) Emergency visit was defined as a case in which a claim was made for emergency medical care after the month following chemotherapy initiation. (2) In Korea, patients are often hospitalized to receive chemotherapy. Therefore, instead of all-cause hospitalization, hospitalization (excluding hospitalization for chemotherapy) after the month following chemotherapy initiation was used as a safety outcome indicator. (3) Febrile neutropenia was defined as hospitalization, claims for short-acting granulocyte colony-stimulating factor (filgrastim, lenograstim), and claims for intravenous antibiotics [35].

### 2.6. Statistical Analyses

Descriptive statistics were used to characterize the study cohort. Categorical variables are presented as number of cases and percentages. Continuous variables are expressed as means and standard deviations (SD) or medians. Age at diagnosis was classified into five groups: 21–50, 51–60, 61–70, 71–80, and ≥81 years. The study was stratified into four periods based on the year of chemotherapy initiation: 2012–2013, 2014–2015, 2016–2017, and 2018–2019.

OS was assessed using the Kaplan–Meier method. The difference between groups was calculated using the log-rank test. The median OS and 1-, 2-, and 3-year survival rates were calculated for each group. The patient death hazard ratios (HRs) were calculated using the Cox proportional hazards model, adjusting for age, sex, CCI score, the year of chemotherapy initiation, and chemotherapy regimens.

We compared the survival outcomes between the GnP and FOLFIRINOX groups using PSM. The χ^2^ test and standardized difference were used to validate significant differences between matched cohorts. Standardized differences between the adjusted covariates were calculated, and differences ≤0.1 were considered to represent an acceptable balance [36]. In the analysis of safety outcomes, the HR between the groups was estimated by Cox regression analysis using the time taken for the first safety event to occur. All statistical tests were two-sided, and statistical significance was set at *p* < 0.05. SAS statistical software version 9.4 (SAS Institute Inc., Cary, NC, USA) was used for all analyses.

## 3. Results

### 3.1. Study Population

A flow chart of the study population is shown in Figure 1. In total, 54,396 patients were registered with PC in the KCCR (2011–2019). Among them, we identified 49,589 patients newly diagnosed with PC based on NHIS data (2012–2019). Patients aged < 20 years (*n* = 69), those with pancreatic neuroendocrine tumors (*n* = 1068), and those with unreliable data (*n* = 1149) were excluded. Consequently, the PC cohort included 47,303 patients (2012–2019). Using the SEER stage of KCCR, we extracted the patients with metastasis (*n* = 21,899). After excluding 1281 patients who underwent surgery and 69 patients who started chemotherapy after 2019, 8652 patients receiving chemotherapy for mPC (2012–2019) were finally included.

The baseline characteristics of the patients are shown in Table 1; 5192 (60.0%) patients were men and 3460 (40.0%) were women. The age group 61–70 years was the largest with 3169 patients (36.6%), and the age group ≥ 81 years was the smallest with 163 patients (1.9%). The most common histological diagnosis was adenocarcinoma (88.5%); 6.5% of the cases were not histologically confirmed. The number of patients in the study period gradually increased over time, with the highest number in 2018–2019. The most common chemotherapy regimen was GnP (2984 patients; 34.5%), followed by gemcitabine plus erlotinib (2099 patients; 24.3%). The median cycle and duration of chemotherapy were 2.3 and 3 months for gemcitabine monotherapy, 3.0 and 3 months for gemcitabine plus erlotinib, 4.7 and 6 months for GnP, and 9.0 and 8 months for FOLFIRINOX, respectively.

### 3.2. Current Status and Survival Outcomes of Chemotherapy for Patients with mPC

#### 3.2.1. Rate of Chemotherapy

The overall chemotherapy rate increased slightly from 40.8% in 2012 to 47.5% in 2019. Chemotherapy rates varied by age group. Chemotherapy was administered to 66.3%, 64.3%, 55.8%, 30.4%, and 5.5% of patients aged 21–50, 51–60, 61–70, 71–80, and ≥81 years, respectively (Appendix A). The most notable increase in chemotherapy rates was observed in patients aged 71–80 years (from 25.5% in 2012 to 40.7% in 2019) (Appendix A).

#### 3.2.2. Types of Chemotherapy

An analysis of the chemotherapy type revealed that gemcitabine plus erlotinib and gemcitabine monotherapy were the main treatments administered from 2012 to 2015. In Korea, insurance coverage has been applied to GnP and FOLFIRINOX since 2016 and 2017, respectively. These two regimens became the standard therapies for mPC after 2017. Among all patients, 85.9% (2018) and 89.0% (2019) received chemotherapy with GnP or FOLFIRINOX. Although the use of FOLFIRINOX gradually increased from 2017 to 2019, GnP was selected more frequently than FOLFIRINOX (Figure 2).

#### 3.2.3. Overall Survival

The OS of patients receiving chemotherapy improved significantly from 2012–2013 to 2018–2019 (median OS: 6, 7, 9, and 10 months in 2012–2013, 2014–2015, 2016–2017, and 2018–2019, respectively; log-rank *p* < 0.0001) (Figure 3A). Since the introduction of GnP and FOLFIRINOX in Korea, OS has improved significantly. Patient OS differences were dependent on the type of chemotherapy administered. The median OS was shorter for gemcitabine monotherapy (6 months) and gemcitabine plus erlotinib (6 months), and longer for GnP (10 months) and FOLFIRINOX (11 months) (log-rank *p* < 0.0001; Figure 3B). The 1-year survival rate was 18.7% for gemcitabine monotherapy, 18.2% for gemcitabine plus erlotinib, 42.3% for GnP, and 48.9% for FOLFIRINOX. The median OS of patients who did not receive chemotherapy was 2 months for all study periods. This observation did not change by year (log-rank *p* = 0.3264; Appendix A). OS showed differences depending on age group. It was longer (9 months) in patients aged 21–50 and 51–60 years, and shorter (6 months) in those aged ≥81 years (log-rank *p* < 0.0001; Appendix A).

#### 3.2.4. HR for Overall Mortality

The HR for overall mortality was analyzed using multivariate Cox regression (Table 2). Women had a better prognosis (HR: 0.931, 95% confidence interval (CI): 0.891–0.972) than did men. When compared with patients aged 21–50 years, there was an increase in HR for those aged 70–79 years (HR: 1.128, 95% CI: 1.134–1.336) and those aged ≥81 years (HR: 1.401, 95% CI: 1.181–1.662). Based on the group with a CCI score ≤5, the HRs of the 6–9 and ≥10 points groups were 1.128 (95% CI: 1.060–1.195) and 1.245 (95% CI: 1.182–1.312), respectively. OS significantly improved in 2018–2019 compared to 2012–2013 (HR: 0.892, 95% CI: 0.815–0.975). Compared to gemcitabine monotherapy, there was an improvement in survival when using GnP (HR: 0.633, 95% CI: 0.580–0.691) and FOLFIRINOX (HR: 0.594, 95% CI: 0.530–0.658).

### 3.3. Comparison between GnP and FOLFIRINOX

#### 3.3.1. Propensity Score Matching

To compare the two standard regimens, GnP (*n* = 2984) and FOLFIRINOX (*n* = 1136), we performed PSM with age, sex, and CCI score. After matching, 1134 patients were selected in both groups and no significant difference in age, sex, or CCI score was observed. An acceptable balance was maintained between the two groups, with a standardized difference of ≤0.1 (Table 3).

#### 3.3.2. Overall Survival after Propensity Score Matching

Before PSM, the median OS significantly differed between the GnP (10 months) and FOLFIRINOX (11 months) groups (*p* = 0.0029; Figure 4A). After PSM, no significant difference was observed between the median OS of the GnP (11 months) and FOLFIRINOX (11 months) groups (*p* = 0.2438; Figure 4B).

#### 3.3.3. Safety Outcomes

After PSM, we analyzed the safety outcomes based on emergency center visits, febrile neutropenia, and hospitalization (excluding hospitalization for chemotherapy). To eliminate the time bias, we analyzed the HR of the first safety indicator experience using the Cox regression method. Compared with the GnP group, the FOLFIRINOX group had a higher HR for febrile neutropenia (HR: 2.285, 95% CI: 1.864–2.802) and hospitalization (HR: 1.16, 95% CI: 1.056–1.274). There were no significant differences in emergency center visits between the two groups (Table 4).

#### 3.3.4. Subgroup Analysis by Age Groups

As a subgroup analysis, we compared the survival outcomes of the GnP and FOLFIRINOX groups by age group after PSM. No significant differences were observed in the survival curves between the two groups in any age group (Appendix A).

## 4. Discussion

This nationwide population-based study analyzed the status of chemotherapy and survival outcomes in patients with mPC using the NHIS–KCCR linked database. Findings from a population-based observational study regarding outcome improvements that are comparable to those of clinical trials may clarify the use of a treatment and support its effectiveness [31]. Herein, we observed an improvement in OS in patients receiving chemotherapy, likely because the standard treatment was changed to GnP and FOLFIRINOX. The median OS increased from 6 months (2012–2013) to 10 months (2018–2019). To our knowledge, this is the largest population-based study to examine the real-world comparative effectiveness of GnP and FOLFIRINOX for patients with PC. After PSM, there was no difference in OS between the two groups. Febrile neutropenia and hospitalization (excluding hospitalization for chemotherapy) occurred less frequently in the GnP group than in the FOLFIRINOX group.

The average chemotherapy rate in patients with mPC was 42.3% and only increased slightly from 40.8% (2012) to 47.5% (2019). This is because the number of elderly patients abstaining from chemotherapy increased. The proportion of patients aged ≥ 81 years receiving chemotherapy remained at 6.4% in 2019. Previous population-based studies reported that among patients with PC aged ≥80 years, 15% in Japan [37] and 21.3% in Canada [10] received chemotherapy. In Spain, only 8% of patients with unresectable PC received chemotherapy [38]. As this study included only patients with mPC, the low chemotherapy rate in those aged ≥ 81 years is similar to that in other countries. However, this study showed that the proportion of patients aged in their 70s who received chemotherapy is gradually increasing. Older patients have various concerns regarding chemotherapy. Therefore, further studies are needed to analyze whether chemotherapy in older patients contributes to improved survival outcomes and to provide clinical evidence.

GnP and FOLFIRINOX have been shown to improve survival outcomes in phase III studies [7,8]. Real-world clinical outcomes may differ owing to differences in physicians, patients, and medical environments in each country [28]. In a population-based study in Ontario, Canada, the median OS of patients who received FOLFIRINOX chemotherapy (2011–2014) was 8.2 months, higher than that of patients receiving gemcitabine monotherapy (4.8 months) [9]. Another Canadian study reported that, compared with 2007, the 1-year survival rate of patients receiving chemotherapy in 2015 improved from 20% to 35%, and the 2-year survival rate improved from 5% to 9% [10]. Recently, we reported on the treatment trends and survival outcomes of all patients with PC in Korea using the NHIS database, and confirmed that the OS in the chemotherapy group gradually improved in 2015–2017 and 2018–2019 [11]. However, because the study had no information regarding the cancer stage, we could not distinguish between locally advanced PC and mPC in the chemotherapy group. Recently, a link between NHIS and KCCR was made possible owing to the policy on the use of healthcare data in Korea, allowing us to obtain cancer stage data. In this study, by selecting only patients with metastases, we compared the survival outcomes of patients receiving chemotherapy at the same stage. The median OSs in the GnP (10 months) and FOLFIRINOX (11 months) groups were similar to those found in previous studies [13,16,39]. Depending on the year, survival significantly improved gradually from 2012–2013 (median OS: 6 months) to 2018–2019 (median OS: 10 months). GnP and FOLFIRINOX were administered to 85–89% of patients receiving chemotherapy in 2018–2019. In this study, we demonstrated that GnP and FOLFIRNOX were sufficiently administered in general patients, resulting in improved survival outcomes for patients with mPC in Korea. The effectiveness of both regimens was confirmed in the general population.

Although GnP and FOLFIRINOX have been shown to improve OS in clinical trials, indirect comparisons between the two regimens are limited owing to differences in study populations and designs. In a retrospective study of patients with mPC in the United States comparing GnP (*n* = 337) and FOLFIRINOX (*n* = 317) groups, the median OS was 12.1 and 13.8 months, respectively, with no difference between the two groups [13]. In a similar study in Austria, the median OS of the GnP (*n* = 297) and FOLFIRINOX (*n* = 158) groups for advanced PC was 10.1 and 11.2 months, respectively, and no significant difference was observed, even when compared by inverse probability of treatment weighting [16]. In Korea, several retrospective studies comparing GnP and FOLFIRINOX have reported different results [14,15,17,20]. In a systematic review of 34 clinical studies, the median OS in patients receiving GnP and FOLFIRINOX for advanced PC was 14.4 and 15.9 months, respectively. More studies have reported a slightly longer median OS for FOLFIRINOX than for GnP; however, these differences were not statistically significant [40]. Meanwhile, according to a recently reported meta-analysis of seven phase III clinical trials, the OS associated with GnP administration (*n* = 1765) was shorter (10.4 months) than that (11.7 months) associated with FOLFIRINOX administration (*n* = 433); however, there was no statistically significant difference (HR: 1.11, (95% CI: 0.88–1.39), *p* = 0.37). In addition, the FOLFIRINOX groups included only patients from two trials aged ≤75 and ≤76 years, respectively, whereas the GnP groups did not have age restrictions in five trials, which represents a difference in patient composition [27].

There are several population-based studies comparing the two regimens. In a Canadian population-based study, the median OS of the FOLFIRINOX group (*n* = 632) was significantly longer than that of the GnP group (*n* = 498; 9.6 vs. 6.1 months) [21]. A similar study in the United States showed that the median OS was approximately 2 months longer in the FOLFIRINOX group (*n* = 566) than in the GnP group (*n* = 536) [22]. In a Dutch population-based study, the median OS for patients with mPC in the FOLFIRINOX group (*n* = 1029) was 6.6 months, which was longer than the 4.7 months in the GnP group (*n* = 207) [24]. Our study included the largest number of patients to date, with 2984 and 1136 patients in the GnP and FOLFIRINOX groups, respectively. Before PSM, the median OS in the FOLFIRINOX group was 11 months, which was significantly longer than that in the GnP group (10 months). After PSM, the median OS in both groups was 11 months, with no significant difference.

The results of this study are notably different from those of previous population-based studies. The first reason is that the medical environment and ethnicity of patients differ from those in western countries. Second, the proportions of patients treated with the two regimens differed. In Europe, first-line treatment preferences vary by country depending on reimbursement and availability [41]. FOLFIRINOX is more frequently used in France and the United Kingdom, whereas GnP is more frequently used in Italy and Spain [41]. In The Netherlands, FOLFIRINOX was most used for mPC (2015–2018) at 64.4%, and GnP was used in only 8.4% of the cases [42]. In previous population-based studies, the number of patients using FOLFIRINOX was greater than those using GnP, although a difference in the degree was noted [21,22,25]. Patients treated with FOLFIRINOX were reported to be younger and to have a better performance status than those treated with GnP [16,21,22]. In Korea, GnP was more frequently used than FOLFIRINOX. The possible reasons include the following. First, few studies comparing GnP and FOLFIRINOX were published before 2020. Second, since GnP was reimbursed a year earlier, physicians who were familiar with GnP may have preferred it to FOLFIRINOX, which has higher toxicity. Third, in Korea, before 2020, most patients were hospitalized to receive FOLFIRINOX; therefore, some patients may have preferred GnP, which could be administrated as outpatient chemotherapy. The fact that there were more patients in the GnP group than in the FOLFIRINOX group suggests the possibility that GnP was used in more patients with good performance compared with other studies. This might result in more improved OS in the GnP group compared to previous population-based studies, and no difference in OS between the two regimens. Nevertheless, we present data comparing two regimens in clinical practice using nationwide real-world data of 1:1 PSM. Therefore, this study provides an objective basis for the selection of a first-line standard treatment for mPC. Further studies considering the healthcare environment of each country are required to compare the effectiveness of these two standard therapies.

In this study, febrile neutropenia occurred less frequently in the GnP group than in the FOLFIRINOX group, consistent with findings from similar studies [13,14,16,17,21,27,29]. However, more hospitalizations (excluding those for chemotherapy) were observed in the FOLFIRINOX group, which differs from the findings of other population-based studies [21,22]. These results should be interpreted considering the medical environment in Korea, as the convenient access to healthcare and lower cost burden compared with other countries have influenced ER visits and hospitalizations. Nevertheless, when selecting a first-line treatment among two standard regimens, these results are worth considering along with the patient’s age, performance, and comorbidities.

This study had some limitations. First, NHIS and KCCR data have limitations in clinical research use; therefore, data on the size of the tumor, metastasis location, performance status, progression-free survival, and adverse effects could not be obtained. The performance status of a patient may influence the decision on anticancer treatment and the choice of anticancer agent, and can, therefore, affect the survival outcome. Second, to protect personal information, age was provided at 5-year intervals and relevant dates were provided in months. Survival time could, therefore, only be calculated in months; however, as the study was conducted among a large number of patients, this is expected to have little effect on the overall results. Third, because we investigated the survival outcomes based only on first-line chemotherapy, the influence of subsequent chemotherapy was not reflected. In Korea, FOLFIRINOX, GnP, and liposomal irinotecan with fluorouracil and leucovorin were only covered by insurance as second-line chemotherapies after 2021. This was after the study period; therefore, an analysis of the impact of second-line chemotherapy could not be conducted. Population-based studies that consider second-line chemotherapies are expected in the near future.

Nevertheless, this study has the following strengths. First, to the best of our knowledge, it included the largest number of patients with mPC who received chemotherapy among all studies, and it is the first nationwide population-based study, enabling the provision of meaningful real-world evidence. This was possible owing to Korea’s NHIS, in which nearly all Koreans are enrolled. By including all patients with mPC in Korea, we have obtained real-world results. Second, although there was no cancer staging information in the NHIS data, patients with mPC could be selected based on cancer stage by linking the KCCR data. Third, because the number of patients in the GnP group was more than twice that in the FOLFIRINOX group, the CCI score could be used for PSM with age and sex. Adjusting comorbidities to match is a strength that differs from that of other studies. Fourth, although population-based studies evaluating safety outcomes are rare, this study presented objective results on the safety outcomes of chemotherapy using an appropriate operational definition.

## 5. Conclusions

Using the NHIS–KCCR linked database, this nationwide population-based study demonstrated that OS was improved with GnP and FOLFIRINOX treatments in patients receiving chemotherapy for mPC. In the PSM comparative analysis between the GnP and FOLFIRINOX groups, no difference in OS was observed. Febrile neutropenia and hospitalization (excluding hospitalization for chemotherapy) were less frequent in the GnP group than in the FOLFIRINOX group. This real-world evidence may help guide decision-making regarding first-line chemotherapy regimens for mPC. Further study is required to reinforce these findings.

## Figures and Tables

**Figure 1 cancers-16-03173-f001:**
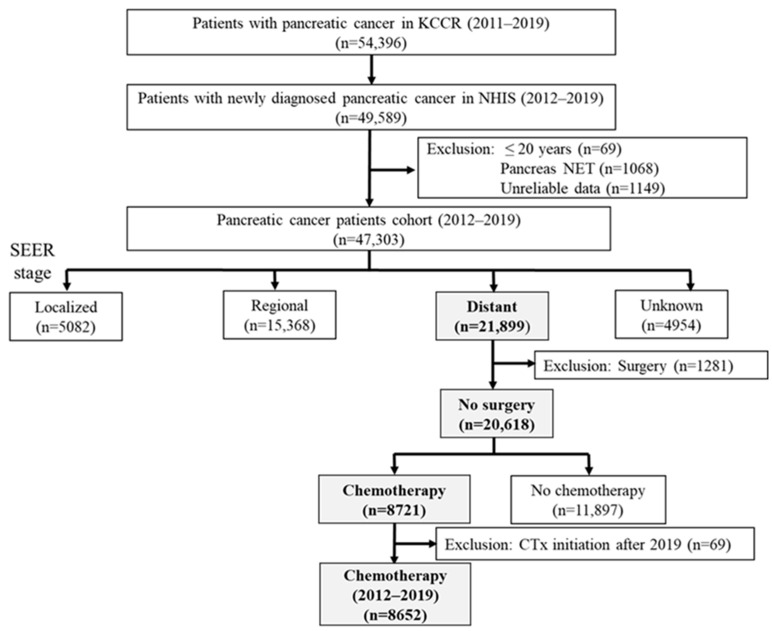
Flow chart of the study population. Abbreviations: NHIS, National Health Insurance Service; KCCR, Korea Central Cancer Registry; NET, neuroendocrine tumor; SEER, the Surveillance, Epidemiology, and End Results Program.

**Figure 2 cancers-16-03173-f002:**
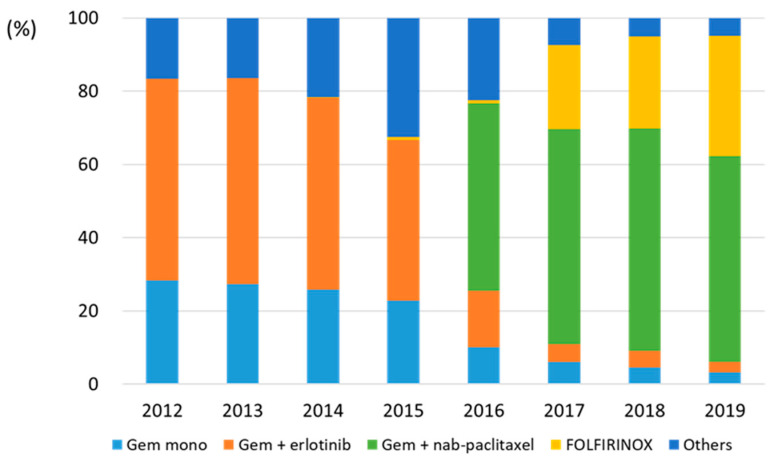
Annual trends of chemotherapy in patients with metastatic pancreatic cancer between 2012 and 2019. Abbreviations: Gem, gemcitabine; Mono, monotherapy.

**Figure 3 cancers-16-03173-f003:**
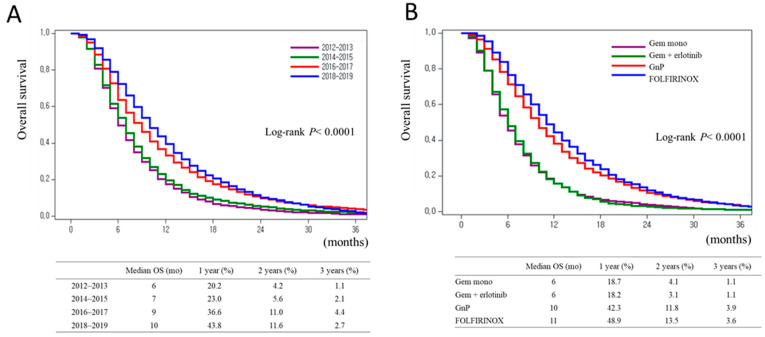
Overall survival of patients with metastatic pancreatic cancer. (**A**) Patients who received chemotherapy by year. (**B**) Patients who received chemotherapy according to chemotherapy type.

**Figure 4 cancers-16-03173-f004:**
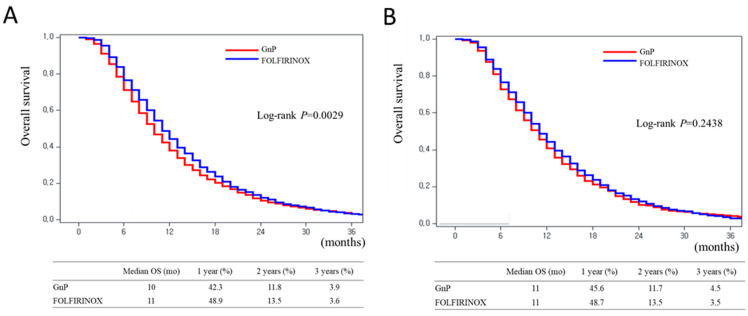
Kaplan–Meier curves of the gemcitabine plus nab-paclitaxel and FOLFIRINOX groups. (**A**) Before propensity score matching. (**B**) After propensity score matching.

**Table 1 cancers-16-03173-t001:** Baseline characteristics of patients with metastatic pancreatic cancer receiving chemotherapy.

Characteristics		No. of Patients	%
Sex	Male	5192	60.0
	Female	3460	40.0
Age (years)	21–50	887	10.3
	51–60	2388	27.8
	61–70	3169	36.6
	71–80	2045	23.6
	≥81	163	1.9
CCI	≤5	2379	27.5
	6–9	2458	28.4
	≥10	3815	44.1
Pathology	Adenocarcinoma	7656	88.5
	Unspecified carcinoma	403	4.7
	Squamous cell carcinoma	23	0.3
	Sarcoma	4	0.0
	Not confirmed	566	6.5
Years	2012–2013	1581	18.3
	2014–2015	1838	21.2
	2016–2017	2372	27.4
	2018–2019	2861	33.1
CTx type	Gem mono	1185	13.7
	Gem + erlotinib	2099	24.3
	GnP	2984	34.5
	FOLFIRINOX	1136	13.1
	Others	1248	14.4
Median CTx cycle (mean ± SD)	Gem mono	2.3 (3.7 ± 3.9)	
	Gem + erlotinib	3.0 (3.8 ± 3.3)	
	GnP	4.7 (5.6 ± 4.3)	
	FOLFIRINOX	9.0 (11.2 ± 8.6)	
Median duration of CTx (months) (mean ± SD)	Gem mono	3 (4.6 ± 5.4)	
	Gem + erlotinib	3 (4.8 ± 5.8)	
	GnP	6 (8.4 ± 7.2)	
	FOLFIRINOX	8 (9.4 ± 7.1)	

Abbreviations: CCI, Charlson comorbidity index; CTx, chemotherapy; SD, standard deviation; Gem, gemcitabine; Mono, monotherapy; GnP, gemcitabine plus nab-paclitaxel.

**Table 2 cancers-16-03173-t002:** Hazard ratio for overall mortality in patients with metastatic pancreatic cancer receiving chemotherapy.

Characteristics		Hazard Ratio	95% CI
Sex	Male	1.0 (ref.)	
	Female	0.931	0.891–0.972
Age (years)	21–50	1.0 (ref.)	
	51–60	0.989	0.915–1.070
	61–70	1.056	0.978–1.140
	71–80	1.231	1.134–1.336
	≥81	1.401	1.181–1.662
CCI	≤5	1.0 (ref.)	
	6–9	1.128	1.066–1.195
	≥10	1.245	1.182–1.312
Years	2012–2013	1.0 (ref.)	
	2014–2015	0.958	0.895–1.025
	2016–2017	0.927	0.854–1.007
	2018–2019	0.892	0.815–0.975
CTx type	Gem mono	1.0 (ref.)	
	Gem + erlotinib	1.026	0.955–1.103
	GnP	0.633	0.580–0.691
	FOLFIRINOX	0.594	0.536–0.658

Abbreviations: CI, confidence interval; ref., reference; CCI, Charlson comorbidity index; CTx, chemotherapy; Gem, gemcitabine; Mono, monotherapy; GnP, gemcitabine plus nab-paclitaxel.

**Table 3 cancers-16-03173-t003:** Baseline characteristics before and after propensity score matching of patients receiving chemotherapy with gemcitabine + nab-paclitaxel or FOLFIRINOX.

Characteristics		Before PSM						After PSM					
		GnP (*n* = 2984)		FOLFIRINOX (*n* = 1136)		*p*-Value	Std. Diff.	GnP (*n* = 1134)		FOLFIRINOX (*n* = 1134)		*p*-Value	Std. Diff.
		N	%	N	%			N	%	N	%		
Sex	Male	1760	59.0	680	59.9	0.0608	0.0179	675	59.5	680	60.0	0.8305	0.009
	Female	1224	41.0	456	40.1			459	40.5	454	40.0		
Age (years)	21–50	243	8.1	146	12.9	<0.0001	0.2988	144	12.7	144	12.7	1	0
	51–60	791	26.5	374	32.9			374	33.0	374	33.0		
	61–70	1167	39.1	419	36.9			419	37.0	419	37.0		
	71–80	746	25.0	192	5.0			192	16.9	192	5.0		
	≥81	37	1.2	5	0.4			5	0.4	5	0.4		
CCI	≤5	864	29.0	424	37.3	<0.0001	0.1801	423	37.3	423	37.3	1	0
	6–9	843	28.3	306	26.9			306	27.0	306	27.0		
	≥10	1277	42.8	406	35.7			405	35.7	405	35.7		

Abbreviations: PSM, propensity-score matching; Std. diff., standardized differences; GnP, gemcitabine plus nab-paclitaxel; CCI, Charlson comorbidity index.

**Table 4 cancers-16-03173-t004:** Hazard ratio of developing safety outcome events.

Outcomes	HR of Developing Events (95% CI)		
	GnP (*n* = 1134)	FOLFIRINOX (*n* = 1134)	*p*-Value
Emergency center visit	1.0 (ref.)	1.046 (0.937–1.168)	0.4221
Febrile neutropenia	1.0 (ref.)	2.285 (1.864–2.802)	<0.0001
Hospitalization *	1.0 (ref.)	1.160 (1.056–1.274)	0.0019

* Excluding hospitalization for chemotherapy. Abbreviations: HR, hazard ratio; CI, confidence interval; GnP, gemcitabine plus nab-paclitaxel; ref., reference.

## Data Availability

The data that support the findings of this study are available from the corresponding author upon reasonable request.

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
