# Peer review of "Real-World Outcomes of First-Line Chemotherapy in Metastatic Pancreatic Cancer: A Nationwide Population-Based Study in Korea"

_cancers, 2024, doi:10.3390/cancers16183173_

Round 1
Reviewer 1 Report
Comments and Suggestions for Authors
1. What is the added value of this data given the multiple retrospective reviews of real-world data in many different populations? The fact that the data is from a very large cohort is good, but are the results providing new knowledge?
2. The population studied includes patients treated with therapies other than the two that the study question is looking at (FFX and GnP). Including data from patients receiving therapies other than these 2 does not add anything to the results, as the question does not concern regimens such as single-agent gemcitabine, or gemcitabine and erlotinib. Including data from patients treated before FFX and GnP were available is not helpful. The start date should be from when both regimens were available to the Korean population for patients with PDAC. Including data from treatments other than FFX and GnP makes the data set look large, but the number treated with FFX or GnP is much less than the total included.
3. There is no value added showing that more patients received treatment by year from 2011 to 2019. The incidence is increasing and older patients are being treated as a routine. I do not see any value in showing that more patients receive treatment during the study period. It is also to be expected as GnP and FFX became available in 2013 and 2011 respectively - although not necessarily funded in Korea in those years.
4. The most common diagnosis was adenocarcinoma of the pancreas (PDAC), 88.5%. Is there any value including patients with other histology? The data presented is intended to compare FFX with GnP in patients with PDAC, and no other patients should be included.
5. The fact that data was not available to show which patients received second-line therapy makes an overall survival comparison less valid. The sequence of therapies may make a difference to survival if both FFX and GnP are used in any particular patient. Patients who did not tolerate one regimen or the other may have switched after a few treatments, confounding the overall survival estimate. These need to be accounted for in a survival assessment.
6. The discussion needs to explain the importance of the findings, not just list them. There needs to be some interpretation of the data. For example, patients who received FFX had less ER visits, but were admitted to hospital more often. How do patients get admitted in Korea? Do they not present to the ER and then get admitted, or is it more common to admit from clinic? There needs to be more interpretation other than to say that some data is consistent with other data published - what do the results mean?
7. The discussion paragraph on the limitations of the study needs to address the details above. It also needs to comment further on how the limitations might affect the data presented. There were many details that could not be determined from the data set used, such as ECOG, that greatly influences the choice of regimen, and reflects how sick they might be, which affects the survival outcome.
Other recommendations:
For the introduction, there is a growing body of clinical trial evidence exploring the question being addressed in this paper. Real-world evidence has its advantages, but also limitations as it is retrospective. The trial evidence has the benefit being controlled and collecting a complete set of pre-determined data. For example, the GENERATE trial (ESMO abstract 16160 2023 comparing FFX with GnP (and S-IROX) in a multicentre randomized phase II/III. There is data showing that NALIRIFOX and FFX are similar in benefit, and both better than GnP when used as a comparitor (ie. NAPOLI3, and the recent JAMA Network Open systemic review and meta-analysis comparing NALIRIFOX, FFX, and GnP as first line therapy (2024)). There is available and up to date data providing quality values for the question being asked in this article. It would be helpful to discuss what the authors feel the advantages of real-world data are over what is emerging now.
The figures need to be improved and made clearer. Figure 2 does not provide any important data. The question relevant is why is GnP being used more than FFX when both are available? The ECOG would help explain that if available, but Figure 2 describes the data gathered, but it isnt answering any relevant question. Figure 3 shows data for patients who received chemotherapy. It is not important to the question being asked. 3A shows the survival over years, of course people will do better when better treatments become available. 3B shows patients who did not receive chemotherapy - they all did poorly as the lines overlap - as would be expected. But adds nothing to answer the question. 3C says patients who received chemo by year, but it shows by age group. 3D includes regimens other than FFX and GnP, and this isnt relevant to the question being asked.
Finally, I am not clear on the value of using CCI - not something routinely used in practice, at least it does not have more weight than the patient's performance status. For example, a patient with a high CCI may have an ECOG of 1, whereas someone with a lower CCI score may have an ECOG of 3 because of the cancer. In routine practice, the performance status often dictates who can receive therapy, and helps determine who might tolerate FFX or GnP. As the population ages, the CCI will increase, but that does not really seem to stop us from treating good-functioning patients.
Reviewer 2 Report
Comments and Suggestions for Authors
Review of paper: "Real-world Outcomes of First-line Chemotherapy in Metastatic Pancreatic Cancer: A Nationwide Population-based Study in Korea"
Strengths:
- The study uses a large and comprehensive dataset from the National Health Insurance Service (NHIS) and the Korea Central Cancer Registry (KCCR), which includes data from over 98% of the Korean population. This makes the findings broadly applicable.
- Rather than using controlled clinical trials, the research offers real-world data, which is helpful for understanding how therapies function in typical clinical situations. Gemcitabine plus nab-paclitaxel (GnP) and FOLFIRINOX provide a more realistic picture of their efficacy and safety.
- Propensity Score Matching (PSM) is used to carefully account for patient characteristics such as age, sex, and general health status, which improves the accuracy of the comparison between GnP and FOLFIRINOX.
- The study provides an in-depth review of overall survival, demonstrating how new chemotherapy therapies have improved survival over time. This serves to emphasize the advancements in the management of metastatic pancreatic cancer.
- A balanced understanding of the advantages and disadvantages of each treatment is provided by including safety outcomes, such as hospitalization and febrile neutropenia rates, which is crucial for making well-informed treatment decisions.
Recommendations:
- Since these can influence treatment outcomes, the paper would be stronger if it included more specific information about the patients' demographics and health characteristics, such as their socioeconomic level, lifestyle, and other medical issues.
- There is room for improvement in the methodology section, e.g., when it comes to describing the steps involved in Propensity Score Matching, the variables that were taken into account, and how the matching enhanced the comparison.
- A more comprehensive assessment of the outcomes might be possible if additional variables, such as the cancer's stage, the patient's general condition, and prior treatments, were included in the analysis.
- A discussion of any variations in the application of GnP and FOLFIRINOX in various Korean hospitals and locations would be beneficial. This could make it easier to comprehend why some therapies could be more popular or accessible in some places.
- Future study topics, such as the long-term effects of these medicines or the potential effects of second-line therapy on patient outcomes, should be suggested by the authors. Investigating genetic and molecular markers may also make it easier to customize care for specific individuals.
Overall Assessment: The paper offers useful insight into the treatment of metastatic pancreatic cancer, which is a significant contribution. The dependability of the results is increased by using meticulous statistical techniques and a sizable, representative dataset. By taking into account the suggestions, the work might be improved even more, providing more in-depth analysis and directing future studies in this crucial field.
Round 2
Reviewer 1 Report
Comments and Suggestions for Authors
Thank you for considering my comments. The majority of my concerns remain. As an oncologist outside of Korea, the data presented will not impact my selection of therapy beyond what evidence is already available. Much of the data is historical and does not reflect current practice.
Round 3
Reviewer 1 Report
Comments and Suggestions for Authors
Overall:
As an oncologist who treats pancreatic cancer, I still do not see any benefit including historical data. Figure 2 does not provide me with any information that is helpful in determining which of the 2 main treatments is superior if either. The bar graphs from 2012-2016 are not relevant to the question oncologists are asking. When comparing this study to others, you only highlight FOLFIRNOX v GnP, because that is the real question being reviewed. Figures 3A-C can be supplemental. For 2D, the inclusion of historical therapies not used anymore is not helpful. It has been known for many years that gem-erlotinib has minimal benefit (albeit statistically significant) over gemcitabine - not clinically significant. To keep in the gemcitabine alone is a good comparison as has been used in trials. It can be a bit of control for the data on FOLFIRINOX and GnP.
In the discussion, the preferential use of GnP and the rationale for that were appropriately provided. However, there is no comment on what the impact may be on the survival outcomes being presented. As some patients or oncologists might have chosen GnP for various reasons (ie. avoiding hospitalization, or familiarity with GnP), what is the impact on the data analysis? For example, if a healthy patient chose GnP for convenience, they may have done better on FOLFIRINOX, but by receiving GnP, it made the GnP look better. If enough patients did this, then many fit patients might have done even better on FOLFIRINOX - and the data might has shown superiority of FOLFIRINOX. This is a limitation of the data, not a comment on the data gathered or the analytic technique.
The comment that this is the largest study is true, but again, it includes so many patients treated before GnP and FOLFLIRINOX were available that it looks much bigger than it is (N=8652, when only 4120 are relevant). It is still a large population.
Lastly, using reference 31 to make the argument for real-world data is not necessary. The authors can make the statement without needing something to back it up. There is clinical value in having real-world data, as is mentioned, trial patients are often different than real world patients.
